# Molecular Hydrogen for Outpatients with COVID-19 (Hydro-COVID): A Phase 3 Randomised, Triple-Blinded, Pragmatic, Placebo-Controlled, Multicentre Trial

**DOI:** 10.3390/jcm13154308

**Published:** 2024-07-24

**Authors:** Yoann Gaboreau, Aleksandra Milovančev, Carole Rolland, Claire Eychenne, Jean-Pierre Alcaraz, Cordelia Ihl, Roseline Mazet, François Boucher, Celine Vermorel, Sergej M. Ostojic, Jean-Christian Borel, Philippe Cinquin, Jean-Luc Bosson

**Affiliations:** 1CNRS, UMR 5525, VetAgro Sup, Grenoble INP, CHU Grenoble Alpes, TIMC, UMR5525, University Grenoble Alpes, 38700 Grenoble, France; carole.rolland@univ-grenoble-alpes.fr (C.R.); claire.eychenne@univ-grenoble-alpes.fr (C.E.); jean-pierre.alcaraz@univ-grenoble-alpes.fr (J.-P.A.); cihl@chu-grenoble.fr (C.I.); francois.boucher@univ-grenoble-alpes.fr (F.B.); celine.vermorel@univ-grenoble-alpes.fr (C.V.); jean-luc.bosson@univ-grenoble-alpes.fr (J.-L.B.); 2Institute of Sremska Kamenica, Cardiovascular Diseases of Vojvodina, 21204 Sremska Kamenica, Serbia; aleksandra.milovancev@mf.uns.ac.rs; 3CHU Grenoble Alpes, Department of Pharmacy, University Grenoble Alpes, 38700 Grenoble, France; rmazet@chu-grenoble.fr; 4FSPE Applied Bioenergetics Lab, University of Novi Sad, 21000 Novi Sad, Serbia; sergej.ostojic@chess.edu.rs; 5Agiradom, 38400 Meylan, France; j.borel@agiradom.com; 6CHU Grenoble Alpes, CIC1406, University Grenoble Alpes, Inserm, 38700 Grenoble, France

**Keywords:** COVID-19, molecular hydrogen, nutrient, primary health care, outcome assessment

## Abstract

**Background**. Due to its antioxidant, anti-inflammatory, anti-apoptosis, and anti-fatigue properties, molecular hydrogen (H_2_) is potentially a novel therapeutic nutrient for patients with coronavirus acute disease 2019 (COVID-19). We determined the efficacy and safety profile of hydrogen-rich water (HRW) to reduce the risk of COVID-19 progression. **Methods**: We also conducted a phase 3, triple-blind, randomised, placebo-controlled trial to evaluate treatment with HRW initiated within 5 days after the onset of signs or symptoms in primary care patients with mild-to-moderate, laboratory-confirmed COVID-19. Participants were randomised to receive HRW or placebo twice daily for 21 days. The incidence of clinical worsening and adverse events were the primary endpoints. **Results**: A total of 675 participants were followed up to day 30. HRW was not superior to placebo in preventing clinical worsening at day 14: in H_2_ group, 46.1% in the H_2_ group, 43.5% in the placebo group, hazard ratio 1.09, 90% confidence interval [0.90–1.31]. One death was reported at day 30 in the H_2_ group and two in the placebo group at day 30. Adverse events were reported in 91 (27%) and 89 (26.2%) participants, respectively. **Conclusions**: HRW taken twice daily from the onset of COVID-19 symptoms for 21 days did not reduce clinical worsening.

## 1. Introduction

To date, more than 800 million cases of coronavirus disease 2019 (COVID-19) have been confirmed, with a notable increase in the Omicron variant during December 2021. More than 7 million deaths worldwide have been reported due to severe acute respiratory syndrome coronavirus 2 (SARS-CoV-2) [1]. A small percentage of COVID-19 patients required hospitalisation, mostly older adults and people with pre-existing conditions (e.g., obesity, diabetes mellitus, hypertension, or disability conditions) [2,3,4]. To prevent these serious complications, several vaccines have been developed in record time. These are highly effective in reducing COVID-19-related hospitalisations, intensive care unit admissions, and deaths [5]. Although two-thirds of the world’s population had been vaccinated with at least one dose and one-third with a booster dose by early December 2023, vaccine coverage is still low and access is uneven [6]. At the same time, antiviral therapies are emerging to reduce the risk of severe COVID-19. Molnupinavir, nirmatrelvir, and ritonavir have been introduced in some countries since 2021 [7,8]. Their introduction in primary care still faces several obstacles: availability in different countries, access to the population, restrictions on their use, conditions of use in the first days of the disease, and a cost-effectiveness ratio that remains to be proven. Other therapeutic approaches have been explored, such as the re-use of molecules such as chloroquine, ivermectin, doxycycline, azythromycin, colchicine, or vitamin D. Nutrition and supplementation were also explored for the prevention, management, and recovery of COVID-19 [9]. All of these strategies have failed to reduce the risk of developing a severe form of COVID-19 in the primary care setting [10,11,12,13,14,15,16]. Inhaled budesonide may provide a small benefit in high-risk patients, but only one serious trial suggests this, and these data need to be confirmed in a larger population [17]. Molecular hydrogen (H_2_) represents a novel approach. Potential preventive and therapeutic applications of H_2_ in various acute and chronic clinical conditions have been strongly suggested [18,19,20,21]. Thus, H_2_ could be a possible adjunctive therapy, especially in COVID-19, to combat an excessive proinflammatory response and, in particular, increased oxidative stress and apoptosis, due to its anti-inflammatory and antioxidant properties [18,22,23,24,25]. Recent studies have confirmed the benefit of H_2_/O_2_ mixing in severe COVID-19 to limit complications or in combination with a rehabilitation programme in post-acute COVID-19 [26,27,28]. By far the simplest and most practical method of H_2_ administration is the oral ingestion of a novel dietary supplement, Hydrogen Rich Water (HRW), which has been widely used with a very low rate of side effects [19,29,30,31]. Moreover, the US-FDA considered HRW GRAS (Generally Recognised As Safe). Since H_2_ has been shown to be effective against pro-inflammatory agents, it could reduce the destructive cytokine storm caused by SARS-CoV-2 at an early stage [32]. We hypothesise that the administration of HRW during the very first days of the disease, at the stage of mild or moderate ambulatory COVID-19, may prevent the inflammatory cascade leading to the cytokine storm and the dramatic consequences of severe COVID-19. Maintaining H_2_ intake during the 3 weeks of COVID-19 could limit the disabling symptoms of the acute phase, such as dyspnea and fatigue, and the progression to severe symptoms.

## 2. Materials and Methods

### 2.1. Study Design

A phase 3 double-blinded, parallel-group, randomised, placebo-controlled trial (RCT) was initiated in January, 2021 to evaluate the safety and efficacy of H_2_ in COVID-19 disease in adult outpatients.

This trial was conducted in 5 French and 1 Serbian region. It was coordinated by the TIMC public laboratory (Grenoble, France). The trial was conducted according to the principles of the International Conference on Harmonisation of Good Clinical Practice guidelines.

In France, an enrollment visit was arranged by video teleconsultation or, alternatively, by telephone with a trained physician investigator. In Serbia, enrollment and product delivery were carried out directly by the investigators during the medical visit at the general practitioner’s office. Sociodemographic characteristics and comorbidities were collected at baseline. Data were collected every day during the first month on a paper CRF at home by participants. The primary outcome was collected on days 12–14 by video teleconsultation, telephone, or during the doctor’s visit by the investigators, depending on the health organisation in the two countries. Secondary outcomes were collected at 1, 3, and 12 months by postal questionnaire and telephone call by the research team. All adverse events were reported. 

### 2.2. Participants

Outpatients with mild biologically confirmed COVID-19, according to World Health Organisation (WHO) guidelines, and with at least one risk factor were eligible, regardless of whether they had been vaccinated against SARS-CoV-2. Inclusion and exclusion criteria are detailed in the Appendix A.

### 2.3. Randomisation and Masking

Eligible and consenting subjects were then randomised in a double-blind fashion using computer-generated random numbers in a 1:1 ratio to either the intervention (HRW) or placebo group. Randomisation was stratified in blocks of four stratified by age (<70 or ≥70 years). Both the HRW pill and the placebo were effervescent pills packaged in identically shaped bottles to maintain blindness. The trial team, investigators, and participants are not informed of treatment allocation until all participants have completed the one-year follow-up visit.

### 2.4. Interventions

All participants received the usual standard of care for COVID-19 provided by their general practitioner. High-concentration HRW was provided via H_2_-producing tablets (Drink HRW). Participants took 1 tablet twice daily in 250 mL of water. The placebo contained identical ingredients to the hydrogen supplement, but instead of metallic magnesium, the placebo contained magnesium carbonate.

### 2.5. Outcomes

The primary endpoint is a composite endpoint of symptom worsening (dyspnea and fatigue), Oxygen 2 loading at home or in an emergency department, hospitalisation (not just emergency department use), and death occurring within 14 days of enrolment. Secondary outcomes included time to clinical improvement, number of days with dyspnea or fatigue, time to hospitalisation for any cause or due to COVID-19 progression, all-cause mortality and time to death from any cause, quality of life and quality of sleep, adverse effects of study medication, and the proportion of participants who were non-adherent. All secondary outcomes were assessed up to one month after randomisation.

### 2.6. Safety 

Safety endpoints included all adverse events occurring during the treatment period (from day 28 or earlier), serious adverse events, and adverse events leading to discontinuation of treatment or placebo. 

### 2.7. Statistical Analysis

Quantitative variables were described as medians and quartiles; qualitative variables were described as frequencies and percentages. Time-to-event variables were described in the same way as qualitative variables.

The primary analysis compared the proportions of patients in the two groups measuring the efficacy of H_2_ compared to placebo. This was assessed by the first primary efficacy endpoint recording up to day 14, using the Kaplan-Meier method to account for all patients, including those who withdrew prematurely from the trial or were lost to follow-up. The adjusted hazard ratio was calculated, stratified by age 70, and associated with 90 confidence intervals for analysis using the Cox proportional model. Results for physical fatigue, psychological symptoms, breathlessness, hospitalization, oxygen therapy, and death separately were presented identically.

For secondary outcomes, treatment adherence and persistent symptoms were compared between groups using the Fisher’s exact test. The time to symptoms resolution between groups was analysed using the Mann-Whitney test. The EQ5D5L was analysed using a linear mixed effects model to model all times, using the individual as a random effect and an interaction between the time (discrete) and the randomisation group. In the same way, the PSQI was analysed using a linear mixed effects model. Subgroup analyses were performed by adding an interaction term between the treatment and the parameter in the Cox model for the main analysis.

The planned enrolment of 700 participants was selected to ensure greater than 95% power to demonstrate superiority in the primary endpoint at a one-sided 2.5% alpha level if the underlying event rates were 24% with HRW and 32% with placebo. All analyses were performed using STATA, version 17 (StataCorp, College Station, TX, USA).

Interim analyses were not planned. We planned a re-evaluation of the event rate when clinical worsening could be evaluated for approximately 100 patients. We planned to stop the study if the event rate was too low, increasing the sample size beyond feasibility. 

## 3. Results

### 3.1. Participants 

Between 22 January 2021 and 24 March 2022, a total of 700 patients were enrolled at baseline and were randomised into the H_2_ group or placebo group. Twenty-five patients were excluded, evenly distributed between the two groups (Figure 1). Finally, 675 patients were analysed at the follow-up period (day 30). Baseline demographic and clinical characteristics were generally similar between the two groups (Table 1). Overall, 69.5% of participants had the onset of signs or symptoms 3 days or less prior to randomization; 71.7% were vaccinated against SARS-CoV-2; and half (53%) were infected with the Delta variant; and 38% with Omicron. The most common risk factors were age over 60 years (41.6%) and obesity (28.6%). Survival status was confirmed for all at day 30.

### 3.2. Efficacy

Hydrogen was not superior to placebo at day 14 in preventing clinical worsening. In the H_2_ group, 154/334 participants (46.1%) experienced clinical worsening as defined by the primary endpoint, compared with 145/333 (43.5%) in the placebo group, hazard ratio: 1.09, 90% confidence interval [0.90–1.31], *p*: 0.479 (Figure 2). Of the clinical criteria making up the primary composite criterion: physical or mental fatigue, breathlessness, need for oxygen therapy, hospitalization, or death, none of them was improved by hydrogen compared with placebo (Table 2). A subgroup analysis was performed to examine the interaction between the main criterion and age, type of SARS-CoV-2 variant, vaccination status, or country of enrollment. No statistical interaction was observed. A sensitivity analysis was performed, excluding the participant who experienced clinical worsening on the first day after randomisation. In this case, 126/306 participants (41.2%) in the H_2_ group and 116/304 (38.2%) in the placebo group experienced clinical deterioration, *p* = 0.389.

The large majority of hospitalisations occurred during the first 2 weeks, and all 3 deaths were attributed to COVID-19-related respiratory distress. Quality of life improved during the first month of follow-up (relative difference of 20.7%), but in a strictly parallel manner between the two groups. The time to resolution of symptoms was 11 days [IQR: 7–17] in the H_2_ group and 14 days [IQR: 7–20] in the placebo group, *p* = 0.186. At one month of follow-up, 39% of participants had persistent symptoms with no difference between the two groups: anosmia, dysgeusia in 58 (10%), dyspnea in 55 (9.5%), cough in 62 (10.7%), and arthralgia in 65 (10.6%). 

### 3.3. Safety 

The incidence of adverse events occurring during or after the treatment period was similar in the two groups (Table 3). No drug- or placebo-related serious adverse events of grade 5 or 4 were reported during the acute phase. Grade 3 events were more frequent, and 45 of the 48 serious situations were related to the severity of COVID-19 and hospitalisation. A suspected serious, unexpected adverse reaction was observed in one patient after 3 days of treatment: lack of energy, insomnia, visual and hearing hallucinations, anxiety, and lower limb oedema. These symptoms were attributed to the placebo group. 

Ulcerative syndrome and hepatic colic were detected in the H_2_ group. One drug-drug interaction was observed with azythromycin and colchicine, requiring hospitalisation.

H_2_ treatment was well tolerated, with 85.5% of participants completing the 21 day treatment course with good compliance (Table 2). Among the reasons for discontinuation, 10 patients described allergic-type reactions (5 in each group), and 92 patients (44 vs. 48) experienced expected side effects such as diarrhoea or abdominal pain. A smaller proportion of 16 patients experienced vomiting (8 in each group). One patient reported a feeling of high blood pressure and was randomised to the placebo group. A few patients reported a sensation of metal rubbing on certain dentures only in the H_2_ group. Less than 1% of patients reported other Class 1 adverse events. 

This section may be subdivided into subheadings. It should provide a concise and accurate description of the experimental results, their interpretation, and the experimental conclusions that can be drawn.

## 4. Discussion

### 4.1. Summary

These data from the Hydro-Covid phase 3 trial in non-hospitalised at-risk adults with COVID-19 indicate that HRW, initiated within 5 days of symptoms onset, does not reduce the risk of hospitalisation for any cause or death by day 30. The speed of symptom recovery was equivalent in the two arms. All patients had regained a good quality of life and good sleep at day 30, regardless of treatment. The safety profile of HRW was reassuring, with the incidence of AEs in line with expectations.

### 4.2. Strengths and Limitations 

The absence of a detected effect may be attributed to a number of factors, including the use of a composite primary endpoint that was too “soft”. A patient-centred approach was employed, whereby symptoms and discomfort were measured using VAS and validated scales, combined with stricter criteria such as hospitalisation, oxygen therapy, and death. This data is illustrated by the fact that 44.8% of patients met the primary endpoint, which was mainly symptomatic criteria, while 3.3% met severe COVID-19 criteria. The profile of the included patients suggested that they were initially in good health, that they were interested in a dietary supplement as the subject of this trial, and that their exposure to pathologies and risk factors was low. In addition, the environmental context, with the rise of the SARS-CoV-2 vaccination, probably played a supplementary protective role. Indeed, public policies have been implemented with the objective of limiting the impact on the hospital system and its associated burden, with a particular focus on at-risk populations. Patients were recruited for this study during the primary and booster vaccination phases. The protective effect has now been demonstrated, with a reduction in the risk of hospitalisation or death in vaccinated patients by a factor of 3 to 10, depending on the number of associated comorbidities [5,33].

Despite the unfavourable outcomes, and the limitations described above, this study has several methodological advantages: it took place solely in patients’ homes, with recruitment exclusively in primary care. In addition, the study validated the recruitment of patients into a clinical trial by teleconsultation, accompanied by a home visit by mobile teams comprising members of the research team or private nurses. This shows that therapeutic trials in primary care are feasible. The clinical trials during the COVID-19 pandemic have facilitated the transition of research activities out of hospital settings, prompting the development of novel strategies for engaging with patients in their own environment. As an illustrative example, the COVERAGE platform trial involved 213 stakeholders in setting up and running the clinical trial for one investigating centre [34]. Of these, more than half were the mobile teams responsible for the inclusion and home follow-up of the included patients.

### 4.3. Comparison with Existing Literature

Other studies have used different concentrations and delivery methods, such as inhalation. In a study conducted by Lebaron et al., changes in biological parameters were observed after 6 months of exposure to HRW administered three times a day [35]. In a further trial conducted in 2019, the efficacy of ingesting 2 tablets of HRW was evaluated on physical performance. It was noted that there was a reduction in heart rate and respiratory rate without any change in VO2 max [36]. Mikami et al. were able to show the efficacy of taking 500 mL of HRW 30 min before physical effort on an ergometric cycle on VO2max and the Borg scale [37]. Botek et al. were able to measure the effect of inhaled H_2_ on effects in terms of improved physical and respiratory function in acute post-COVID-19 patients [28]. Recently, HRW was proven to alleviate fatigue in acute post-COVID-19 patients [38]. These data support a biological and clinical effect of H_2_, but the effective dose and conditions of use have yet to be determined.

### 4.4. Implications for Research and/or Practice

Molecular hydrogen was recently identified as a potential therapeutic nutrient. In three decades, more than 900 clinical trials were described in the PubMed database. A thematic issue consisting of 19 review articles on H_2_ medicine was recently published [39]. The therapeutic possibilities are numerous, both for sick patients—all the more interesting as the inflammatory component is predominant—and for healthy people. This approach deserves to be brought to the attention of Western research teams, who are poorly represented in this particularly promising field of research. Our study has shown that administration in the form of a dietary supplement (HRW) is perfectly tolerated by patients, making it a potentially very interesting tool for pathologies that can be treated on an outpatient basis.

## Figures and Tables

**Figure 1 jcm-13-04308-f001:**
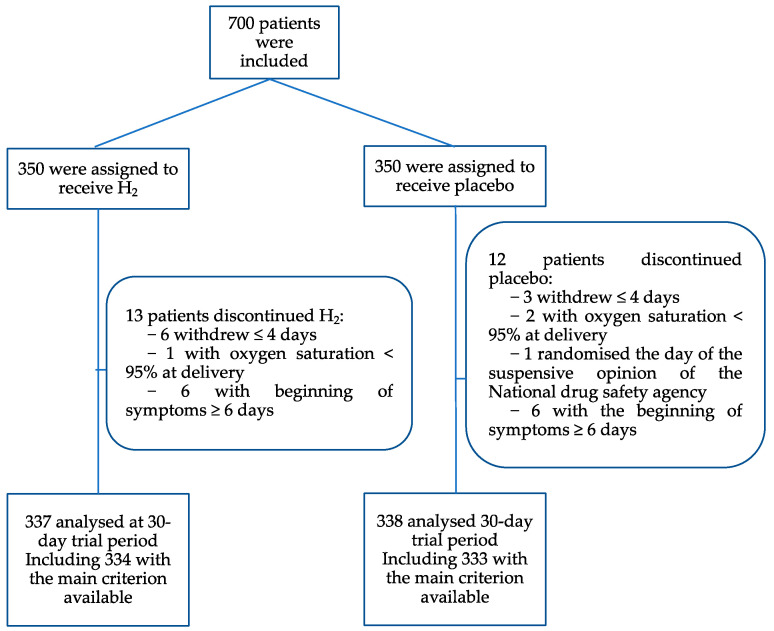
Randomization and Flow of Participants from Baseline.

**Figure 2 jcm-13-04308-f002:**
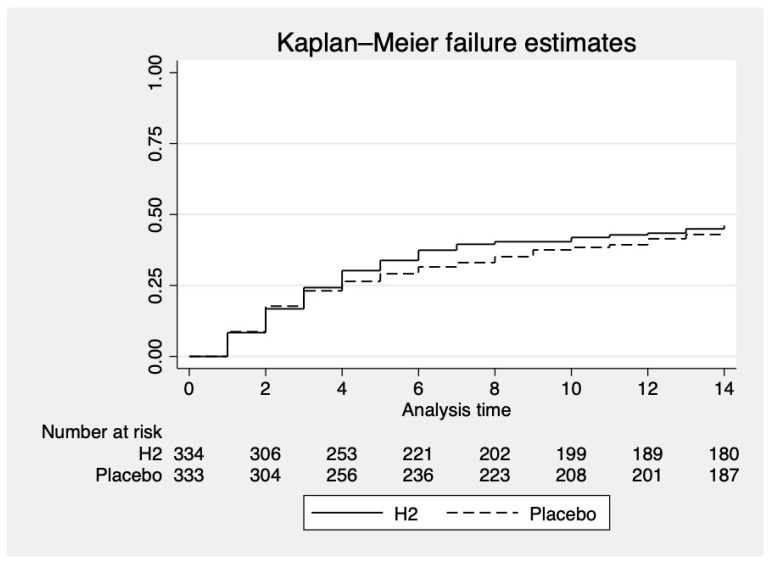
Time-to-Event Analysis of clinical worsening at Day 14.

**Table 1 jcm-13-04308-t001:** Demographic and clinical characteristics of the participants at baseline.

			H_2_ (*n* = 337)	Placebo (*n* = 338)	*p*-Value
Country, no. (%)					0.96
	France		210 (62.3)	210 (62.1)	
	Serbia		127 (37.7)	128 (37.9)	
Median age, [IQR], yr			57 [46–64]	57 [49–64]	0.25
Age group, no (%)					0.11
	<50 yr		114 (33.8)	93 (27.5)	
	50–59 yr		83 (24.6)	110 (32.5)	
	60–69 yr		103 (30.6)	99 (29.3)	
	≥70 yr		37 (11.0)	36 (10.7)	
Sex at birth, no (%)					0.85
	Female		206 (61.1)	209 (61.8)	
	Male		131 (38.9)	129 (38.1)	
Risk factors for severe COVID-19, no. (%)					
	Obesity *		95 (28.2)	98 (29.0)	0.82
		Body mass index > 35 kg/m^2^	27 (8.0)	32 (9.5)	0.5
	Diabetes mellitus		31 (9.2)	35 (10.4)	0.61
	Serious heart condition †		12 (3.6)	13 (3.9)	0.84
	Serious respiratory condition ‡		28 (8.3)	26 (7.7)	0.77
	Chronic kidney disease (Clearance < 60 mL/min/1.73 m^2^)		3 (0.9)	3 (0.9)	1
	Active Cancer		3 (0.9)	4 (1.2)	1
	Association between three non major risk factors		34 (10.1)	36 (10.7)	0.81
Co-existing conditions					
	Tabagism		58 (17.2)	60 (17.8)	0.85
	Hypertension		135 (40.1)	142 (42.0)	0.61
	Dyslipidemia		35 (10.4)	37 (11.0)	0.81
	Asthma		26 (7.7)	22 (6.5)	0.54
Medications at baseline					
	Corticotherapy		13 (3.9)	6 (1.8)	0.1
	Anticoagulation		12 (3.6)	11 (3.3)	0.83
	Non-steroidal anti-inflammatory drugs		31 (9.2)	36 (10.7)	0.53
Quality of life (EQ5D Index score), median [IQR]			0.95 [0.88–0.97]	0.95 [0.9–0.98]	0.99
Quality of life (EQ5-VAS), median [IQR]			65 [50–80]	65 [50–80]	0.83
Quality of sleep (PSQI)			5 [3–8]	5 [3–8]	0.77
Patient vaccinated against SARS-CoV-2, no. (%)			249 (73.9)	235 (69.5)	0.21
Theorical variant designation ¶, *n* (%)					0.69
	Alpha		25 (7.4)	31 (9.2)	
	Delta		182 (54.0)	176 (52.1)	
	Omicron		130 (38.6)	131 (38.8)	
Time since first symptoms to randomization ‖, no. (%)			*n = 316*	*n = 307*	0.65
	≤3 days		217 (68.7)	216 (70.4)	
	>3 days		99 (31.3)	91 (29.6)	
Number of different symptoms in the initial phase, no. (%)					
	1		12 (3.6)	15 (4.4)	0.92
	2		50 (14.8)	51 (15.1)	
	3		71 (21.1)	76 (22.5)	
	4		80 (23.7)	82 (24.3)	
	>4		124 (36.8)	114 (22.7)	

* Obesity was defined by a body-mass index of 30 or higher. † Complicated hypertension, previous stroke, history of coronary artery disease, history of cardiac surgery, heart failure, peripheral arterial disease, atrial fibrillation. ‡ Chronic obstructive pulmonary disease, asthma with inhaled corticosteroid therapy, pulmonary fibrosis, sleep apnea syndrome, cystic fibrosis. ¶ Data were extracted from national biological databases by selecting the most prevalent SARS-CoV-2 variant based on the geographical area and date of inclusion. ‖ The time period was based on data collected at randomisation.

**Table 2 jcm-13-04308-t002:** Primary and secondary outcomes.

			H_2_ (*n*: 334)	Placebo (*n*: 333)	*p*-Value
			Number (percent)		
Primary endpoint *					
	Clinical deterioration at day 14		154 (46.1)	145 (43.5)	0.479
	Physical fatigue		99 (29.4)	88 (26.4)	0.383
		Chalder scale	4 (1.2)	3 (0.9)	
		On a visual analogic scale	37 (11.1)	31 (9.3)	
		Described by the patient	68 (20.4)	61 (18.3)	
	Mental symptoms		46 (13.8)	54 (16.2)	0.391
		Chalder scale	5 (1.5)	7 (2.1)	
		On a visual analogic scale	42 (12.6)	47 (14.1)	
	Breathlessness		78 (23.4)	67 (23.1)	0.31
		On mMRC	17 (5.1)	14 (4.2)	
		On a visual analogic scale	44 (13.2)	43 (12.9)	
		Described by the patient	32 (9.6)	23 (6.9)	
	Hospitalisation/Oxygen therapy required		11 (3.3)	10 (3.0)	0.83
	Death		1 (0.3)	0	
Secondary end points					
Age (yr)					0.6
<50			48 (42.5)	36 (40.9)	
50–59			45 (54.9)	51 (46.4)	
60–69			39 (38.2)	41 (41.4)	
≥70			22 (59.5)	17 (47.2)	
Sex at birth, no (%)					0.73
Female			107 (52.5)	99 (48.3)	
Male			47 (36.2)	46 (35.9)	
	Hospitalisation/Oxygen therapy required at Day-30		12 (2.6)	10 (3.0)	0.83
	Death at Day-30		1 (0.3)	2 (0.6)	0.58
	Treatment compliance at day 14				
		≥80%	252 (76.8)	263 (80.7)	0.252
		≥50%	272 (82.9)	284 (87.1)	0.154
	Treatment compliance at day 21		*n*: 229	*n*: 248	
		≥80%	197 (86.0)	211 (85.1)	0.796
		≥50%	217 (94.8)	234 (94.4)	1
	Quality of life (index score), at day-30 median [IQR]		0.98 (0.93–0.1)	0.98 (0.93–0.1)	0.212
	Quality of life (health status) at day-30 median [IQR]		80 (70–90)	80 (70–90)	0.76
	Quality of sleep, median [IQR]		5 (3–8.7)	5 (3.5–7)	0.047

* Worsening of fatigue was defined by a 25% increase via the Chalder scale (i.e., an increase ≥5 points for physical symptoms and ≥3 points for mental symptoms) or via daily VAS self-assessment for fatigue (i.e., an increase ≥2.5 points). Worsening of dyspnoea was defined by a 25% increase of the mMRC scale (i.e., an increase ≥1 point if mMRC at baseline ≥1 or an increase ≥2 points if mMRC at baseline = 0) or a daily VAS self-assessment (i.e., an increase ≥2.5 points).

**Table 3 jcm-13-04308-t003:** Summary of Adverse Events, Serious Adverse Events, and Adverse Events Leading to Discontinuation through Day 30 *.

Adverse Event Category		H_2_ (*n* = 337)	Placebo (*n* = 338)	*p*-Value
Events that emerged during treatment period				
	Patients with adverse events—no. (%)	91 (27.0)	89 (26.2)	0.84
	Any adverse event	126 (37.4)	124 (36.7)	0.85
	Serious adverse event	21 (6.2)	27 (8.0)	0.79
	Maximum grade 3 or 4 adverse events	7 (2.1)	11 (3.3)	0.34
	Maximum grade 5 adverse event	1	2	1
	Discontinued drug or placebo because of an adverse event	16 (4.7)	14 (4.1)	0.85
	Had a dose reduction or temporary discontinuation owing to an adverse event	2	3	1
Events considered to be related to drug or placebo				
	Patients with adverse events—no. (%)	60 (17.8)	59 (17.5)	0.9
	Any adverse event	76 (22.6)	74 (21.9)	0.84
	Serious adverse event	1	0	1
	Maximum grade 3 or 4 adverse events	1	0	1
	Maximum grade 5 adverse event	0	0	1
	Discontinued drug or placebo because of an adverse event	4 (1.2)	5 (1.5)	0.6
	Had a dose reduction or temporary discontinuation owing to an adverse event	23 (6.8)	21 (6.2)	0.75

* Shown are data for all patients who received at least one dose of drug or placebo. All reported deaths were related to COVID-19; causes of death included COVID-19 pneumonia (3 patients).

## Data Availability

The raw data supporting the conclusions of this article will be made available by the authors on request.

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
