# Peer review of "Molecular Hydrogen for Outpatients with COVID-19 (Hydro-COVID): A Phase 3 Randomised, Triple-Blinded, Pragmatic, Placebo-Controlled, Multicentre Trial"

_jcm, 2024, doi:10.3390/jcm13154308_

Round 1
Reviewer 1 Report
Comments and Suggestions for Authors
Dear Authors;
This study provides Phase 3 trial results for the therapeutic research on COVID-19. I would like to offer some review comments.
Table 1 indicates risk factors for severe illness from COVID-19 such as obesity and diabetes mellitus; however, these are interpreted as baseline health conditions of the participants. These are not risk factors originating from COVID-19 itself, so please revise the phrasing accordingly.
Please provide the statistical significance (p-value) between the H2 group and the Placebo group in Table 3.
The main results from the Hydro-Covid phase 3 trial in non-hospitalized at-risk adults indicate that HRW initiated within 5 days of symptom onset does not reduce the risk of hospitalization or death from any cause through day-30. Detailed analyses revealed no significant differences between the groups. It is recommended, though not mandatory, to provide additional detailed analyses based on demographic characteristics such as age and gender.
The overall quality of the English language is good. However, please double-check for grammatical errors in preparation for the final review.
Author Response
|
1. Summary |
|
|
|
|
Thank you very much for taking the time to review this manuscript. Please find the detailed responses below and the corresponding revisions in the re-submitted files.
|
|||
|
2. Point-by-point response to Comments and Suggestions for Authors 2.1. Reviewer 1 |
|
||
|
Comments 1: Table 1 indicates risk factors for severe illness from COVID-19 such as obesity and diabetes mellitus; however, these are interpreted as baseline health conditions of the participants. These are not risk factors originating from COVID-19 itself, so please revise the phrasing accordingly.
|
|
||
|
Response 1: Thank you for pointing this out. Therefore, we have changes the phrasing in table 1 as “Risk factors for severe COVID-19”
|
|
||
|
Comments 2: Please provide the statistical significance (p-value) between the H2 group and the Placebo group in Table 3.
|
|
||
|
Response 2: We have, accordingly to your suggestion provide p-value in the table 3. Thanks.
Comment 3: The main results from the Hydro-Covid phase 3 trial in non-hospitalized at-risk adults indicate that HRW initiated within 5 days of symptom onset does not reduce the risk of hospitalization or death from any cause through day-30. Detailed analyses revealed no significant differences between the groups. It is recommended, though not mandatory, to provide additional detailed analyses based on demographic characteristics such as age and gender.
Response 3: Thanks for your comment. We add these 2 additional variables along with the p-value data for age and sex in Table 2.
|
|
||
|
4. Response to Comments on the Quality of English Language |
|
||
|
Point 1: The overall quality of the English language is good. However, please double-check for grammatical errors in preparation for the final review
Response 4 : according to your suggestion, we corrected wording in the manuscript.
|
|
||
Reviewer 2 Report
Comments and Suggestions for Authors
Estimated Authors,
I've read with interest the present study on a phase 3 randomised, triple-blinded, pragmatic, placebo-controlled, multicentre trial on the potential clinical effects of hydrogen-rich water supplementation for outpatients with Covid-19. The study, encompassing a total of 700 patients (with a limited dropout rate) substantially failed in identifying a positive role of this intervention in reducing the length and the severity of COVID-19 associated signs/symptoms.
The study has been properly designed, and despite the negative results it provides, has a certain interest for professionals involved in the managing of COVID-19 cases.
From my point of view, only minor adjustments are required, and more precisely:
1) Table 1: please provide univariate p value for cases vs. controls;
2) Table 3: please make the decimal notation consistent, not only in table 3 but also across the main text;
3) It is quite unclear how patients were ultimately recruited (Sections 2.2-2.3). Please provide some further details on the recruitment procedures in order to explain whether included patients could be acknowledged as representative of the general population.
Author Response
Thank you very much for taking the time to review this manuscript. Please find the detailed responses below and the corresponding revisions in the re-submitted files.
2.1. Reviewer 2
Comment 1: Table 1: please provide univariate p value for cases vs. controls;
Response 1: Thanks for your remark. We put in the table 1 univariate analysis for cases vs. control. We hope this will suit you.
Comment 2: Table 3: please make the decimal notation consistent, not only in table 3 but also across the main text;
Response 2: We have, accordingly to your suggestion provide decimal notation consistent in tables and the main text.
Comment 3: It is quite unclear how patients were ultimately recruited (Sections 2.2-2.3). Please provide some further details on the recruitment procedures in order to explain whether included patients could be acknowledged as representative of the general population.
Response 3: thank you so much for this remark. Our methodology was quite complex and too long to explain in this manuscript. In this way, we propose detailed material and methods in supplementary material. For this section, we precise:
Participants
Partnerships were organised with nasopharyngeal swabbing teams and laboratories analysing SARS-CoV-2 retro-transcription – polymerase chain reaction (RT-PCR), general practitioners offices, home-care nurses, pharmacies and vaccination centres. Several community outreach strategies were used, including physical (posters, leaflets, newspapers) and social (radio, networks) media, to publicise the trial. At the beginning of the study, through communication tools or encouragement from their health professional, patients were invited to call the investigation centre to propose their participation. Four months later, due to a low inclusion rate, French medical analysis laboratories in the recruitment area were invited to contact by telephone each patient who tested positive for COVID-19 PCR. If the patient expressed interest, the trial team was informed and in turn provided trial information to the patient and proceeded with pre-inclusion. An inclusion visit by video teleconsultation, or, alternatively, by telephone with a trained physician investigator was arranged, the data were recorded in an electronic case report form (CRF). Products delivery and paper CRF were carried out at patient's home immediately after its inclusion by research team, a medical doctor or a home care nurse partner. Oxygen status was controlled at home before products delivery with a pulse oximeter (superior or equal 95 % to be included) and left at the disposal of the patient to complete paper CRF during follow up. In order to increase the inclusion rate, which has been slowed down by the different epidemic waves, Serbia has been associated to this RCT. Eligible Serbian patients started to be screened in May 2021. Due to the differences in health organisation in France and Serbia, the inclusion and product delivery were carried out in Serbia directly by the investigators during the medical visit.
Inclusion criteria at randomization were as follows: SARS-CoV-2 biologically confirmed for less than 4 days by antigen test or RT-PCR, onset of signs or symptoms for less than 4 days earlier, at least one sign or symptom of COVID-19, and at least one risk factor for development of severe COVID-19 (age >60 years; treated hypertension; obesity; all types of treated diabetes; serious heart conditions [heart failure, coronary artery disease, or atrial fibrillation]; history of stroke; stage 3 chronic renal failure [30 ≤ estimated GFR <60 mL / min / 1.73 m²]; chronic obstructive pulmonary disease including chronic respiratory failure under long-term oxygen therapy; active cancer or one diagnosed in the last 5 years; immunodeficiency of therapeutic origin or HIV infection and last known CD4 count <200 / mm3; history of pulmonary embolism and / or proximal deep vein thrombosis; asthma under inhaled corticosteroid therapy; paired sleep apnoea syndrome; peripheral arterial disease of the lower limbs stage II and above; another risk factor presented, according to the list defined by the French High Council of Public Health.43 Patients with three minor cumulative conditions (i.e. overweight, social vulnerabilities, hypothyroidism, immune or not listed inflammatory disease) could also be eligible according to the investigator. At the beginning of the epidemic, many patients were frustrated by the lack of access to biological tests, as the sampling laboratories were saturated. In the case of absence of biologically-confirmed nasopharyngeal swab, a patient was eligible if he/she presented at least 3 clinical signs among 11 (fever > 37.5°C for at least 3 days; cough; sore throat/cold; headache; anosmia, dysgeusia; myalgias, arthralgias, bone pain; breathing difficulties [feeling of dyspnoea at rest]; chest pain [sternal]; digestive complaints [diarrhoea, nausea, vomiting]; tachycardia [palpitation]; conjunctivitis [red eyes]), and notion of contact (with a COVID+ certain or probable patient) within the last 10 days. In this situation, patients could be included under the condition to obtain a positive result during the following days.
Exclusion criteria were a negative RT-PCR test, absence of attending or referring physician, oxygen saturation < 95% at baseline, any sign of seriousness incompatible with home care, severe chronic renal failure or requiring dialysis (i.e. eGFR <30), uncontrolled clinically significant heart disease, pregnancy, patient under guardianship or curatorship. Any treatment necessary for the management of acute or chronic patient conditions was allowed, (including for example any monoclonal antibody, anti-viral drugs, vitamin D, ivermectin, zinc, colchicine … ). All subjects were asked to maintain the same lifestyle throughout the study. Written informed consent was obtained from all patients.
Randomisation and masking
Baseline data was collected using an on-line case-report form that included demographics, major comorbidities, any treatment or dietary supplement taken by enrolled patients during the study. Eligible and consenting subjects were then randomized in a double-blind fashion to either intervention (HRW) or placebo group by computer-generated random numbers in an 1:1 ratio. Randomisation was stratified in blocks of four stratified by age (< 70 or ≥ 70 years).
Hydro-Covid was initiated on January 22, 2021, when the first participant was screened. The last participant was enrolled between on March 24, 2022, and completed the one-month visit between on April 25, 2022. The trial team, investigators and participants are not informed of treatment allocation until all participants have completed the one-year follow-up visit. The HRW pill and the placebo were packaged in identically shaped bottles. An unblinding procedure has been foreseen in case of necessity (severe allergy, life-threatening emergency) at the request of any physician and validated by the trial team.